# Fiscal policy and economic growth in Sub-Saharan Africa: Do governance indicators matter?

**Isubalew Daba Ayana**[1]☯*, **Wondaferahu Mulugeta Demissie**[2]☯, **Atnafu Gebremeskel Sore**[3]☯

**1** Department of Economics, Wollega University, Nekemte, Ethiopia, **2** Department of Economics, Ethiopian Civil Service University, Addis Ababa, Ethiopia, **3** Departments of Economics, Addis Ababa University, Addis Ababa, Ethiopia

☯ These authors contributed equally to this work.
* isubalewmsc@gmail.com

**Data Availability Statement:** The data underlying the results presented in the study are available from world development indicator (https://databank.worldbank.org/reports.aspx?source=World-Development-Indicators) and World governance indicators (https://databank.

## Abstract

This study investigated the linkage between fiscal policy-governance indicators interaction and economic growth in 36 Sub-Saharan Africa (SSA) countries from the periods of 2011–2021 inclusive. The study employed two-step system Generalized Method of Moment (GMM) estimation technique due to its practical relevance in panel data analysis. The data obtained from World Bank and World governance indicator was checked for unit root through the help of Im Pesaran Shin and Levin-Lin-Chu unit-root tests, and the result revealed that data was stationary and safe for further analysis. The result of the study also presented that direct economic effect of fiscal policy is negative and significant in SSA countries. However, the interaction of fiscal policy with governance indicators has positive and significant effect on economic growth. Accordingly, before interacting with governance indicators, a percentage change in fiscal policy leads to a 0.20 percent decline in economic growth of SSA countries. Contrary to this, the interactive coefficient of fiscal policy and government effectiveness (0.019) and interactive coefficient of fiscal policy and corruption control (0.0046) are found to be positive and significant. Further, the finding of the study revealed that fiscal policy-voice and accountability interaction coefficient (0.011) and interactive coefficient of fiscal policy-regulatory qualities (0.014) are positively and significantly affecting economic growth of SSA countries. The policy implication is that policy makers in SSA countries should encourage economic policies that improve government effectiveness, strong corruption control, clean public services and better regulatory qualities.

## 1. Introduction

Fiscal policy is the most widely experienced macroeconomic policy in the world. It is a tool of achieving economic stability. This viewpoint of fiscal policy was widely accepted across the globe. The other way of inspecting fiscal policy is directed to the amalgamation of its types and objectives. Accordingly, the fiscal stance of a country can be contractionary fiscal policy, which is largely aimed to reduce the aggregate demand in an economy. On the contrary, the main

worldbank.org/source/worldwide-governance-
indicators).

**Funding:** The author(s) received no specific
funding for this work.

**Competing interests:** The authors have declared
that no competing interests exist.

aim of expansionary fiscal policy is intensifications of the aggregate demand in an economy [1, 2].

As fiscal policy is the instrument of economic stability that is implemented by the government, its interaction with the level of governance has been one of the focus areas of empirical research in economics. It was part of academic and policy debate since the time of Keynes which marked the necessity of government intervention to bring growth and stability through deliberate policy tools [3–5].

In the Sub-Saharan Africa region, where there are forty-eight countries over a billion populations are living, the linkage between fiscal policy and governance indicators interaction raises an immense question that needs a detail scientific investigation. The questions in this area can generated from three major reasons. First, the extent of government quality and government effectiveness is lower compared to other parts of the globe. Secondly, the greater corruption index and the lower voice and accountability are particular issues in SSA countries relative to other parts of the world. Finally, questions of economic policy credibility and policy implementation strategies are also more stimulating problems in SSA countries [6, 7].

Several studies are conducted on the relationship between fiscal policy and economic growth. For instance, [8] has investigated the effect of fiscal policy on GDP growth of Sub-Saharan Africa but the emphasis of the study was to boldly show the effect of natural resources rent revenue on GDP. Similarly, [9] examined the link between fiscal policy and economic growth in sub-Saharan African countries. However, the relative importance of the fiscal and monetary policy was focused instead of brashly telling the effect of fiscal policy variables in the area.

The other studies on the relationship between fiscal policy and economic growth are at the individual country level and cannot give us the general picture of the effect of fiscal policy on economic growth in sub-Saharan Africa. Let us show some of the studies and their contradicting findings across the region. As a sample only, a study in South Africa by [10, 11] has shown that the effect of fiscal policy on the economy of the country depends highly on the nature of the policy instruments implemented. This was recently assured, after ten years, through the study of [12].

Despite many studies carried out on the linkage between fiscal policy and economic growth, the literature**s** in SSA countries remained limited. Thus, the main objective of this study is to investigate the effect of fiscal policy and economic growth by governance indicator variables in the model. This study differs from the aforementioned studies in three ways. First, unlike most of the preceding studies, this study relatively covered large cross-sections of Sub-Saharan Africa. Secondly, it introduces governance indicator variables in the model. Thirdly, it employs the contemporary estimation method dubbed as system GMM. Further, this study utilized two-step system GMM as it enables to obtain more efficient estimates compared to one-step system GMM.

Moreover, except [2, 8, 9, 13] no studies, to the best of investigators knowledge, addressed the linkage between fiscal policy and economic growth in sub-Saharan Africa. Further, except the study [14–16] who investigated the direct effect of governance on economic growth of SSA African countries, no empirical works, to the best of author's knowledge, examined the effect of interaction of fiscal policy and governance indicators on economic growth of the region. Thus, this study investigates the effect of fiscal policy and governance indicators interaction on economic growth of SSA countries from the periods of 2011 to 2021 inclusive. Regionally, the scope of this study is limited to Sub-Saharan Africa. In terms of country, it covered only 36 SSA countries due to the problem of data availability. From the angle of time, our study covered eleven years data, almost second decade of 21st century.

This study has a paramount significance in SSA countries as it contributes to existing literatures in the areas of fiscal policy, governance indicators and economic growth. The findings of this study also support the decision making process of the SSA policy makers.

The rest sections of the study are organized as follows. Section 2 presented literature review while methodology and data of the study are conferred in section 3. Further, Section 4 of this study provided data analysis whereas conclusions are discussed in section 5.

## 2. Literature review

### 2.1. Review of related theoretical literature

The evolution and development of fiscal policy date back to the time of the world great depression of the 1930s. After the laissez-faire policy of classical economics is challenged, the debate on whether government involvement in an economic issue was the major agenda of debate among policymakers. The huge devastation faced by the world economy during the time forced the emergence of fiscal policy [17].

Next, the extent of government intervention and the nature of government involvement becomes the major point of discussion in an economy. Classical Keynesian economics was one of the indications of this. Since then the debate on the components and methods of fiscal policy was attracting the attention of the economists studying in this area [18].

Schools of economic thought have their view regarding the framework of economic activities. For instance, classical fiscal policy depends on the idea they are having regarding population, value, interest, money, and unemployment. Moreover, their fiscal policy idea shows their idea on investment, consumption, government expenditure, and government involvement in the economy as well as their idea on saving and financial issues and institutions. The policy of self-inters and the principles of the invisible hand by Smith were considered the central theme of fiscal policy of classical during the time. The classical schools have only one component of fiscal policy. The invisible hand belief covered by the laissez-faire strategy was the fiscal policy of classical [19].

The embryo of Keynesian fiscal policy stands on its assumptions. One issue is regarding the fact that denies the flexibility of wages and prices. Keynesianism believes that the change in collective demand does not influence the level of prices. According to the Keynesian fiscal policy, it is demand that forces the growth of the economy. This process of growth, therefore, is to be sponsored by the fiscal policy. The fiscal policy Keynes was thinking of is mainly of two categories [20].

Institutionalism's view of fiscal policy started being the point of debate just after the term institutional economics was coined by Williamson and immediately taken by well-known economist Rudolf Richter. Starting from that time, institutional economics and its theoretical development along with institutional policy became a point of debate and discussion among economists. They believe that the better the institution, the easier will be the fiscal policy implementation [21].

### 2.2. Review of related empirical literature

Recent empirical findings are thoroughly reviewed in this section. First, the linkage between government expenditure and economic growth is reviewed. Secondly, the empirical literatures that link governance indicators and economic growth are reviewed. Finally, the current empirical finds that connect labor and capital with economic growth were reviewed.

Several studies were conducted on the relationship between fiscal policy and economic growth in Sub-Saharan Africa [9]. Examined the effect of fiscal and monetary policy on the economic growth of using annual panel data of sub-Saharan African countries from 1985 and

2019. Through the use of the dynamic panel VEC model, their study found that fiscal policy is affecting the economic growth of these countries positively.

Moreover, by combining both fiscal and monetary policies, their study revealed that the former is more powerful than the latter in the sub-Saharan African countries under investigation. This fact was recognized by economists such as Robinson, Rodriguez, and Dornbusch in the previous decade. Today, international organizations including IMF and World Bank have also recognized fiscal policy as a tool of stabilizing, designing, and building economy of developing nations in general [7].

Recently, similar work was done by [13], who carried out an empirical investigation of the effect of fiscal policy on GDP growth in Sub-Saharan Africa but failed to capture many of the countries of the region. Their study emphasized Nigeria and Ghana and thus one cannot have the full picture of the line between fiscal policy and economic growth in sub-Saharan African countries. Previously, [2] examined the fiscal policy and economic development of Sub-Saharan Africa but with a particular focus on Ghana and Nigeria too.

[22] Examined the effect of public spending on the economic growth of economic communities of West African countries using panel data analysis. From Hsiao tests of coefficient equality and Panel co-integration models conducted, the study found that total public spending is not positively contributed to the economic growth of majorities of ECOWAS countries in both the short-run and long-run during the period under investigation. Furthermore, the study found that public consumption is also not positively contributing to the economic growth of these countries as depicted from the GMM model applied in the study.

[23] Found that gross domestic investment is the main variable that affects the economic growth of Asian countries by using panel data of 19 countries from 2002–2017 to estimate it through the difference general method of the moment model. Further, their study has shown that current government expenditure reduces the economic growth of these countries during the period under investigation. From the panel granger causality model, their study found that government expenditure on education granger causes economic growth of the countries. But it is found that government expenditure on education fails to contribute significantly to the economic growth of the countries.

[24, 25] Found that public expenditure has an insignificant positive impact on the economic growth of West African Economic and Monetary Union member states using panel data from 1985 to 2011. From the autoregressive model conducted, the study concluded that public expenditure on education and health is categorized as unproductive expenditure in all WEAMU countries. Moreover, the elasticity of the public expenditure was found to be negative in the sample countries. It is also found that the effect of public expenditure on the GDP growth of the nations is different even among the WEAMU member states.

[26] Found a strong long-run relationship between government expenditure and economic growth of lower-income countries in sub-Saharan African regions using panel data from 1980 to 2015 as GMM and statistic panel data estimation models were employed in their study. Their work also indicated that government expenditure and economic growth of the region are negatively related as depicted in Arellano and Bond's general method of moment estimation techniques. Their study remained to worry about the proper disaggregation of the total public expenditure upon prioritization which is highly affected by the fiscal policy of the respective nations.

[27] Investigated the effect of fiscal policy on the economic growth of Asian countries using the auto distributive lag model for panel data between 1970–2016 covering forty-six years of five panels of Asian countries. Their study used human capital, debt, and foreign direct investment as explanatory variables that regressed over the economic growth of the countries. Moreover, their study concluded that human capital is positively contributing to the economic

growth of Singapore while it is supporting the economic growth of Malaysia. Debt is significantly contributing to the GDP growth of Thailand, Indonesia, and the Philippines.

[28] Examined the relationship between government expenditure and economic growth for emerging developing countries using more sophisticated panel data of 59 nations from the period of 1990–2019. From the fixed effect panel estimation technique, their study concluded that government expenditure is positively affecting the economic growth of the developing nations included as a sample in the particular study. Their study supported Keynesian theory which advocates the rise in government expenditure to accelerate the economic growth of these nations. The study of [29] found that expenditure on education and health has positive effect on economic growth of SSA countries during 2006 to 2018. From the two-step system GMM estimation technique, their study revealed that the African countries should direct their expenditure towards health and education instead of military expenditure.

The linkage between capital formation and economic growth has been studied by many scholars. For example, [30] found that there is negative effect of capital formation on economic growth in Africa, Asia and America between 1980–2018. However, the effect of gross capital formation is found to be dependent on the level of income of countries. In Africa, gross capital formation positively affects economic growth. In Africa, their study found that there is causality between gross capital formation and economic growth is unidirectional running gross capital formation to GDP.

[31] Found that growth capital formation is has two purposes in SSA countries. First, it reduces poverty to enhance economic growth. Secondly, gross capital formation enhances institutional development in the region that in turn positively enhances economic growth. Similar finding was previously reported by [32], who investigated the linkage between gross capital formation and economic growth using panel co-integration [33]. Found that capital formation enhances economic growth for the study period of 1970–2012 in India. Their study suggested that government needs to raise capital formation to enhance growth.

[34, 35] Found that effect of capital on economic growth depends on whether the capital stock is owned by the public or private. Using panel data for Latin American countries from the period of 1970 to 2014, their study found public capital crowd private capital in the short run. The result of their study has revealed the positive effect of public capital on economic growth. This implied that the countries should give attention for both private and public investment.

The link between labor and economic growth is studied by many studies. For instance, [36] examined the gap between labor productivity and employment gap in SSA countries. The study found that labor allocation and labor structure in SSA are the major determinants of employment and in turn economic growth. Further, labor is also the center of agricultural productivity that is the central issue in economic growth of SSA countries.

[33] Found that the effect of population on economic growth is based on the level of income of the countries. Accordingly, low population is a problem in high income country while high population is another puzzle in low income countries. Using the historical data over the past 200 years, the study found that the imbalance between income and population contributed to global economic inequality and injustice.

Similar study of [37] found that labor is part of the human capital. Thus, the extent of human capital formation determines the contribution of labor to economic growth. The linkage between labor and economic growth remain positive in rich countries where education and health investments on labor are advanced. Contrary to this, the linkage between labor and economic growth remain negative in low income countries where human capital formation through education is low.

[38] Examined the linkage between population growth and economic development through the use of literature survey. The study found that the nature of economic development determine population growth and development as well as the labor contribution to economy. Further, the nature of population policy also determines the extent of labor utilization in the country.

## 2.3. Governance indicators and economic growth nexus

[3] Investigated the linkage between governance and economic growth in developing countries using the data from 1996 to 2018. Their study found that worldwide governance indicators are contributing differently in different countries. From the fixed effect model analysis, their study revealed that control of corruption is found to significantly contribute to economic growth. Their study also concluded that there is no common consensus among both theoretical and empirical literatures.

[39, 40] Found that the interaction of governance indicators and foreign direct investment enhances economic growth of OECD Countries from the period of 1996–2013 inclusive. From the generalized method of moment applied to the study, their study found that governance indicators and foreign direct investment significantly affect growth of the region while effective government policy and improved regulatory quality were one of the engines of economic growth of SSA countries. Similarly, [41] found that government effectiveness had a positive and significant effect on corruption for the panel of five South Asian countries during the sample study periods of 2003–2018.

[15] Found that corruption control and governance effectiveness is found to have negative and significant effect on economic growth performance during the study periods 2002 to 2020 for 22 SSA countries. On the other hand, his study has shown that regulatory quality has positive effect on economic growth performance of the countries. Similar study was conducted early by [14] investigated the effect of government expenditure and efficiency on economic growth from 2002 to 2015 for low income SSA countries to find that the rising government expenditure promotes economic growth while no evidence was observed when efficiency is interacted with government expenditure.

[42] Found no relationship between the governance indicators and economic growth of countries SSA countries using system generalized method of moment [43]. Found that institutional governance interaction with population turned the negative effect of it to positive for developing and developing 91 countries from the period of 1999 to 2014. Form the random effect model estimation technique conducted, their study found that economic growth though population channel therefore is very important for economic growth. Similarly, [44] found that the governance indicators such as rule of law, corruption control and voice and accountability are positive and significant impact on economic growth using the dynamic pane data from the period of 2002–2018 for 31 developed countries of the world.

Similarly, [45] found that governance and economic growth are positively linked to each other in 116 countries. The positive influence of governance is significant at one percent significant level. The study of [46] concluded that voice and accountability as governance indicator has positive and significant impact on economic growth among east African member countries.

The work of [47] also found that voice and accountability enhance democracy and then economic growth [39, 48]. Investigated the linkage between governance and economic growth for the quarters of the years from 2002 to 2018. Their study found that increasing corruption discourages economic growth while government effectiveness enhances economic growth during the period under investigation. From panel of 14 countries and pooled mean group

estimation, their study recommended that corruption should be reduced and government effectiveness needs to be strengthened.

## 3. Methodology and data

### 3.1. Data and variables of the study

This study employed a panel data of 36 Sub-Saharan African countries for the periods of 2011–2021 to investigate the relationship between fiscal policy-governance interaction and economic growth. Data for the study were sourced from the world development indicator (WDI) and world governance indicator (WGI). The sources were preferred as the institutions provide dependable and reliable data [49, 50].

Table 1 presents variables included in the study and their corresponding expected sign. The governance indicators were selected based on the work of [51–53] whereas labor (Labor (LBF) total) and capital (GFCF) were incorporated in the study following neoclassical economic growth model.

The study employed system generalized method of moment estimation technique as recommended by [54–56]. The estimation technique designed by them is super over other methods due to three facts. First, the method takes the problem of weak instrumentation into account especially during the first difference equation. Secondly, system GMM is very compatible with larger cross sections and short time dimensions (N>T) cases of the dynamic panel

**Table 1. Descriptions of the study variables and expected sign of coefficients.**

| Variables of the study | Description the variables | Expected sign |
|---|---|---|
| Economic growth(GDPR) | GDP per capita is gross domestic product divided by midyear population. It is inflation adjusted GDP. | |
| Government expenditure(GE) as a percentage of GDP | Includes all government current expenditures for purchases of goods and services (including compensation of employees). It also includes most expenditure on national defense and security, but excludes government military expenditures | – |
| Capital (GFCF) as percentage of GDP | Includes land improvements including schools, offices, hospitals, private residential dwellings, and commercial and industrial buildings. It is gross fixed capital formation. | + |
| Labor (LBF) total | Comprises people ages 15 and older who supply labor for the production of goods and services during a specified period. | + |
| Government effectiveness (goef) | Measures the quality of public services, the quality of the civil service and the quality of policy formulation and implementation, and the credibility of the government's commitment to its stated policies. Scored as from -2.5 (less effective) to 2.5 (more effective). | +/- |
| Corruption control (cocor) | Control of Corruption index covers the idea of how public power is exercised for private gain. Scores range from − 2.5 to 2.5, with a higher value indicating that the perception of anti-corruption in the citizens is strong. | +/- |
| Voice and accountability(voac) | The voice and accountability index evaluates the level of freedom a country's citizens have in the expression, association, and other media. It also presents how well citizens are able to participate in their government. Scores range from − 2.5 to 2.5, with higher score indicating a higher relative freedom of speech. | +/- |
| Regulatory quality(requ) | Captures perceptions of the ability of the government to formulate and implement sound policies and regulations that permit and promote private sector development. Performance score from 0 to 100. The highest score reflects the best situation. | +/- |

*Source*: Authors building *Note*: economic growth (GDPR) is the dependent variable of the study

data. Thirdly, system GMM solves the endogeneity and reverse causality problems that are associated to the omitted variable cases. Fourthly, system GMM control for time and individual specific level effects which leads to efficient estimates when compared to two stage least square. This study conducted a two system GMM estimation technique due to the fact that it results in efficient estimates than the one step assuming homoscedasticity.

The Hansen test was employed to test for over identification restriction while Sargan test was employed to check for auto correlation technique. The other issue associated to estimation technique is that the estimation is conducted with the robust standard error as it maintains consistency with the panel specific level auto correlations [57, 58]. Proceeding to estimation, the study conducted unit root tests to check stationary data. Estimation was carried out through statistical software package dubbed as stata15.

## 3.2. System GMM model specifications

In order to understand the effect of fiscal policy-governance interaction on economic growth, we introduced the world wide governance indicators, adopted as recommended by [51–53], specifically corruption control, government effectiveness, political stability, regulatory quality and voice and accountability. The governance indicator added to the model at creating interaction between the indicators and economic growth [59]. In order to study the effect of fiscal policy on economic growth of thirty six SSA countries for the period 2011–2021, we adopted the dynamic panel data model as follows.

$$GDPR_{it} = \text{\th}_0 + \text{\th}_1 GDPR_{t-1} + \text{\th}_2 GE_{it} + \text{\th}_3 \text{Є}_{it} + I_{it} + \varphi_{it} + u_{it} \tag{1}$$

In Eq 1, $GDPR_{it}$ shows real gross domestic product growth, $GDPR_{t-1}$ denoted the first lag of the real GDP, $GE_{it}$ shows general government final consumption expenditure which is the proxy of fiscal policy, $\text{Є}_{it}$ denotes the vectors of control variables in the model. The explanatory variables were included in the model in line with the neoclassical production function.

$$GDPR_{it} = \text{\th}_0 + \text{\th}_1 GDPR_{t-1} + \text{\th}_2 GE_{it} + \text{\th}_3 GFCF_{it} + \text{\th}_4 LBF_{it} + I_{it} + \varphi_{it} + u_{it} \tag{2}$$

In Eq 2 the set of control variables were defined. For instance, $GFCF_{it}$ shows the gross fixed capital formation utilized as the proxy of capital, $LBF_{it}$ as the proxy of labor.

Thus, the interactive model of fiscal policy and economic growth is provided as:

$$GDPR_{it} = \text{\th}_0 + \text{\th}_1 GDPR_{t-1} + \text{\th}_2 GE_{it} + \text{\th}_3 GFCF_{it} + \text{\th}_4 LBF_{it} + \text{\th}_5 INTER_{it} + I_{it} + \varphi_{it} + u_{it} \tag{3}$$

Eq 3 shows the INTER denotes the interaction of GE and the governance indicators ($INTER_{it}$). However, ($INTER_{it}$) appear in the model after interacting with the government expenditure (GE).

Eq 4 introduced fiscal policy-governance interaction indicators in the model and it is written as:

$$GDPR_{it} = \text{\th}_0 + \text{\th}_1 GDPR_{t-1} + \text{\th}_2 GE_{it} + \text{\th}_3 GFCF_{it} + \text{\th}_4 LBF_{it} + \text{\th}_5 GEgoef_{it} +$$
$$\text{\th}_6 GEcocor_{it} + \text{\th}_7 GEvoac_{it} + \text{\th}_8 GErequ_{it} + I_{it} + \varphi_{it} + u_{it} \tag{4}$$

Where, $GEgoef$ shows the interaction of proxy of fiscal policy with governance effectiveness, $GEcocor$ denotes interaction of fiscal policy with corruption control, $GEvoac$ shows interaction of fiscal policy with voice and accountability, $GErequ$ shows the interaction of fiscal policy with regulatory qualities. $\text{\th}_0$ shows constant, $\text{\th}_1$, $\text{\th}_2$, $\text{\th}_3$, $\text{\th}_4$, $\text{\th}_5$, $\text{\th}_6$, $\text{\th}_7$ and $\text{\th}_8$ are the coefficients

of explanatory variables. The governance indicator variables where included in the study based the work of [53].

Eq 5 shows the estimated model in the form of natural logarithm for convenience to get elasticity interpretation of results. Finally estimated model of the study is written as:

$$logGDPR_{it} = þ_0 + þ_1 logGDPR_{t-1} + þ_2 logGE_{it} + þ_3 logGFCF_{it} + þ_4 logLBF_{it} + þ_5 logGEgoef_{it} + \\ þ_6 logGEcocor_{it} + þ_7 logGEvoac_{it} + þ_8 logGErequ_{it} + l_{it} + \varphi_{it} + u_{it}$$

(5)

Where, log is the natural logarithm and all others are explained above.

## 4. Data analysis

### 4.1. Descriptive analysis

We start this section by presenting descriptive aspect of the data. Table 2 presents the mean, standard deviation, maximum and minimum value of the study variables. Accordingly, real GDP is appeared with the mean value of 4.044056 indicating that the average real GDP growth in SSA from the period of 2011–2021 is 4.044. The maximum value is 6.058752 while the minimum value is negative 1.939965. This reflects that there is a great disparity between the real GDP growth rates of SSA Africa countries. Further, standard deviation is provided as 0.0854078 reflecting the real GDP growth of SSA countries vary across countries. The overall observation of the panel data is found to be 396 indicating the observation is large enough to conduct analysis and draw conclusion.

The major study variable, government expenditure (*logGE*), has the mean value of 22.48336 while minimum and maximum are 22.42767 and 22.61258 respectively. This points out that the government expenditure of SSA region varies. The standard deviation of *logGE* is found to be 0.0609911 while the average gross fixed capital formation in SSA countries is 23.12411 whereas the mean value of total labor force is 16.11921. This reveals that government expenditure across SSA countries varies significantly.

On the other hand, corruption control index (cocor), one of the governance indicators introduced in the model, is found to have mean value of negative 0.6859054 indicating that corruption control of SSA countries is still in negative. The lowest value of corruption control index according to world development indicator is negative 2.5 [49, 50]. Further, descriptive statistics result shows that the variable is found with minimum value of -1.88736 and maximum value of 0.9665675 demonstrating that the perception towards anti-corruption across SSA countries is different. This is depicted by the larger standard deviation of the variable 0.6018062.

**Table 2. Results of summary statistics for the study variables.**

| Study variables | In shorts | Observation | Mean | Minimum | Maximum | Standard dev. |
|---|---|---|---|---|---|---|
| Real GDP | *GDPR* | 396 | 4.044056 | -1.939965 | 6.058752 | .0854078 |
| Government expenditure | *logGE* | 396 | 22.48336 | 22.42767 | 22.61258 | .0609911 |
| Gross fix. capital formation | *logGFCF* | 396 | 23.12411 | 22.95838 | 23.67441 | .2037563 |
| Labor force | *logLBF* | 396 | 16.11921 | 15.98809 | 16.26294 | .0904699 |
| Corruption control | *cocor* | 396 | -.6859054 | -1.88736 | .9665675 | .6018062 |
| Government effectiveness | *goef* | 396 | -.7820798 | -1.88736 | 1.16092 | .6138295 |
| Voice and accountability | *voac* | 396 | -.6449944 | - 1.892658 | 1.196947 | .5818975 |
| Regulatory quality | *requ* | 396 | -.5767874 | -1.851003 | .9396508 | .6754053 |

**Source**: Own computation from STATA 15. **Notes**: Except real GDP all variables are in log form.

Government effectiveness (*goef*), the other governance indicator included in our model, is found to have the average value of negative 0.7820798 showing that the quality of public and civil in SSA countries are low. According to World development indicator, the values of government effectiveness lie between negative 2.5 (less effective) to 2.5 (more effective). In addition to this, government effectiveness in SSA lies between negative 1.88736 and 1.16092 showing that there is enormous disparity among SSA countries with standard deviation of 0.6138295. The mean value of voice and accountability (*voac*) is found to be negative 0.6449944 with standard deviation of 0.5818975 in SSA during the study period. Scores of voice and accountability range from negative 2.5 to 2.5. According to the World Bank scores with higher score indicates a higher relative freedom of speech. Regulatory quality (*requ*) in SSA is averaged to negative 0.5767874. The minimum and maximum values of the regulatory quality is negative 1.851003 while 0.9396508 respectively indicating large disparity among SSA countries. Performance score of regulatory quality index ranges from 0 to 100 according to World Bank and the highest score reflects the best situation.

Table 3 displays the list of sample countries with highest and lowest government expenditure, the proxy of fiscal policy in this study. Accordingly, Mauritius is the first country followed by Niger and Guinea-Bissau by the size of government expenditure. On the other hand, Namibia and Comoros ranks 4th and 5th respectively. Contrary to this, Ethiopia is one of the countries with the lowest government expenditure out of the sample countries followed by the Gambia and Rwanda. The fourth lowest rank in terms of government expenditure is occupied by Chad while the fifth rank is belongs to Nigeria. The main implication of this is that private sector is relatively better in the countries with lowest government expenditure while private sector is dominated by the public in the countries with the highest government expenditure.

Table 4 presents sample countries with strong and frail corruption control in SSA countries. Corruption control index shows the perception of anti-corruption in the citizens. The index of corruption control lies between -2.5 and 2.5. The highest value shows that citizens have strong anti-corruption perception. Accordingly, the panel data from 2011 to 2021 confirms that Sudan ranks first by having relatively weak corruption control index followed by republic of Congo and Guinea-Bissau. Further, Chad and Burundi, ranks 4th and 5th respectively, are reflecting that public power is less exercised for private gain. Differing to this, the descriptive statistics of this study shows that Benin, Ethiopia, Eswatini, Ghana and Senegal ranks from 1st to 5th respectively by their corruption control index. This shows that power is more exercised for private gain in these countries. It also reflects that citizen's perception towards anti-corruption is very high. However, the mean value of corruption control index in both categories remains negative. This shows that the corruption index is closer to negative 2.5 instead of positive 2.5. This reflects that corruption in SSA is rampant despite huge efforts countries are exerting to combat it.

**Table 3. Five sample countries with highest and lowest government expenditure in SSA.**

| highest government expenditure | | | lowest government expenditure | | |
|---|---|---|---|---|---|
| Rank | Country | Mean of logGE | Rank | Country | Mean of logGE |
| 1 | Mauritius | 22.483357 | 1 | Ethiopia | 22.483357 |
| 2 | Niger | 22.483357 | 2 | Gambia, The | 22.483357 |
| 3 | Guinea-Bissau | 22.483357 | 3 | Rwanda | 22.483357 |
| 4 | Namibia | 22.483357 | 4 | Chad | 22.483357 |
| 5 | Comoros | 22.483357 | 5 | Nigeria | 22.483357 |

**Source**: Own computation from STATA 15. **Notes**: government expenditure is in log form.

**Table 4. Five sample countries with lowest and highest mean of corruption control index.**

| Lowest mean of corruption control index | | | Highest mean of corruption control index | | |
|---|---|---|---|---|---|
| Rank | Country | Mean of cocor | Rank | Country | Mean of logGE |
| 1 | Sudan | -1.4368797 | 1 | Benin | -.53246761 |
| 2 | Congo, Dem. Rep. | -1.4256318 | 2 | Ethiopia | -.48314511 |
| 3 | Guinea-Bissau | - 1.4244886 | 3 | Eswatini | -.24696951 |
| 4 | Chad | -1.39343 | 4 | Ghana | -.1301765 |
| 5 | Burundi | -1.3879236 | 5 | Senegal | -.07914409 |

**Source**: Own computation from STATA 15. **Notes**: cocor shows control of corruption.

## 4.2. Econometric analysis: Unit root test results

The unit root test by Im, Pesaran, and Shin was employed as it captures the limitations posed by the LLC unit root test. It is argued by [60] that the IPS is less restrictive in assumptions. Another reason that forces the selection of IPS in the study is that it is more flexible and user-friendly (popular) by way of it takes into account the likelihood method of calculating unit root. This study also conducted one of the first generation unit panel unit root tests, the LLC unit root test as recommended by [61].

Table 5 presents the unit root test results for IPs and LLC. The result reveals that the all variables are stationary at level except *goef* and *requ*. Although it is possible to proceed with stationary variables, the two variables non-stationary at level should be checked for unit root. Thus, we conducted the unit root test of the variables at first difference.

Table 6 presents Im Pesaran Shin (IPS) and Levin-Lin-Chu unit-root tests at first difference. The result of the tests shows that all variables of the study are stationary after converting them to first difference. Thus, the variables of the study are safe and secure for further analysis. This implies that our data is free from spurious regression and the results are reliable.

The study also carried out Hausman test to select appropriate model for the estimation. The result indicated that system GMM is more suitable compared to difference GMM. This implies that the relevant model of estimation specific to the data is employed following Hausman test. Thus, appropriateness of our estimation model reveals that findings and policy implications delivered by this study are reliable [58].

**Table 5. Results of Im Pesaran Shin (IPS) and Levin-Lin-Chu unit-root test at level.**

| Study Variables | Im Pesaran Shin(IPS) unit root test at level | | Levin-Lin-Chu(LLC) unit-root test at level | | |
|---|---|---|---|---|---|
| | Z-t-tilde-bar | p-value | Adjusted t*Statistic | p-value | Decision at I(0) |
| *logGDPR* | -3.5377 | 0.0002** | -21.2155 | 0.0000** | Stationary |
| *logGE* | -3.4348 | 0.0003** | -4.2871 | 0.0000** | Stationary |
| *logGFCF* | -10.2172 | 0.0000** | -7.1475 | 0.0000** | Stationary |
| *logLBF* | -6.6067 | 0.0000** | -13.0400 | 0.0000** | Stationary |
| *cocor* | -3.6234 | 0.0001** | -4.6287 | 0.0000** | Stationary |
| *goef* | -3.2666 | 0.7305*** | -6.8759 | 0.5600*** | Non Stationary |
| *voac* | -5.6575 | 0.0000** | -5.0324 | 0.0000** | Stationary |
| *requ* | -2.2400 | 0.5125*** | -6.3798 | 0.6700*** | Non Stationary |

**Source**: Own computation from STATA 15. **Notes**: ** denotes 1% level of significance respectively while *** shows insignificant variables. All variables are in log form. Cocor is corruption control, goef is government effectiveness, voac is voice and accountability and requ is regulatory quality. They are the proxy of governance indicators.

**Table 6. Results of Im Pesaran Shin (IPS) and Levin-Lin-Chu unit-root tests at I(1).**

| Study Variables | Im Pesaran Shin(IPS) test at first difference | | | Levin-Lin-Chu(LLC) test at first difference | | |
|---|---|---|---|---|---|---|
| | t-tilde-bar | Z-t-tilde-bar | p-value | Adjusted t*Statistic | p-value | Decision at I(1) |
| *logGDPR* | -1.7182 | -3.5377 | 0.000* | -22.4616 | 0.0000* | Stationary |
| *logGE* | -2.4671 | -9.7113 | 0.0000* | -21.6006 | 0.0000* | Stationary |
| *logGFCF* | -1.6818 | -3.2380 | 0.0006* | -13.0986 | 0.0000* | Stationary |
| *logLBF* | -2.5285 | -10.2172 | 0.0000* | -10.4248 | 0.0000* | Stationary |
| *cocor* | -2.1431 | -7.2243 | 0.0000* | -14.5299 | 0.0000* | Stationary |
| *goef* | -1.4548 | -1.3669 | 0.0858*** | -8.2596 | 0.0132** | Stationary |
| *voac* | -2.0768 | -6.6731 | 0.0000* | -9.0314 | 0.0001* | Stationary |
| *requ* | -1.8507 | -4.7936 | 0.0000* | -9.8173 | 0.0000* | Stationary |

**Source**: Own computation from STATA 15. **Notes**:

*, **, and *** denotes 1%, 5% and 10% level of significance respectively. All variables are in log form. Cocor is corruption control, goef is government effectiveness, voac is voice and accountability and requ is regulatory quality. They are the proxy of governance indicators.

Table 7 shows the result of Hausman test for model selection. The estimated result of pooled OLS model is found to be 0.515906 whereas the fixed effect model is 0.507541 less than the pooled model. This implies that fixed effect model estimation technique is more appropriate compared to pooled OLS model. Similarly, the difference GMM model revealed 0.3258245 showing that it is less than that of all. This suggests that system GMM is the appropriate model in estimating the effect of fiscal policy on economic growth of SSA countries. Thus, we employed system GMM to estimate the model.

## 4.3. Discussions

This section discusses results of the two-step system GMM on the link between fiscal policy and economic growth in SSA countries during the time span (2011–2021) under investigation.

Table 8 presents the result of system generalized method of moment estimated using two-step system GMM. The result revealed that the first lag of the dependent variable economic growth is found to be positive. The implication is that contemporaneous growth is positively and significantly affected in SSA countries at 1 percent level of significance, other things remaining constant. This result is uniform in all three columns of the result. This is true and logical as the last year economic growth can enhance the current economic growth. It is this fact that is holding true in SSA countries. This might happen in several channels such as increase in aggregate demand, raising an investment and supplying capital for economic growth. The result also depicted that a percentage increase in lag-one of economic growth leads to 0.526 percent rise in contemporaneous growth in SSA countries over the period under investigation. Our finding regarding this corroborates with the work of [29].

Another point of our finding goes to the effect of general government final consumption expenditure which is the proxy of fiscal policy in our study. The study found that the general final government consumption expenditure has negative and significant effect on economic

**Table 7. Hausman test result for model selection.**

| Lagged dependent variable | Pooled OLS model | Fixed Effect Model | Difference GMM model | Decision |
|---|---|---|---|---|
| L1.logGDPR | 0.515906 | 0.507541 | 0.3258245 | System GMM model is appropriate |

**Source**: Own computation from STATA 15. **Note**: L1.RGDP is the lag 1 of the dependent variable (real GDP) of the study.

**Table 8. Fiscal policy and economic growth relationship in SSA: Do governance indicators matter?**

| Variables | System GMM result(two-step)(1) | Fixed effect model(2) | Random effect model(3) |
|---|---|---|---|
| L1.logGDPR | 0.5266749* | 0.507541* | 0.515906* |
| | (0.0061067) | (0.1180978) | (0.1122481) |
| logGE | -0.2002726 * | -0.2065206** | -0.2094484** |
| | (0.005365) | (0.0245516) | (0.0232885) |
| logGFCF | 0.0386406* | 0.0342265 ** | 0.0350103** |
| | (0.0092529) | (.0231853) | (0.0220682) |
| logLBF | 0.4581424* | 0.4717203* | 0.4650048* |
| | (0.041383) | (0.1098208) | (0.1044911) |
| GEgoef | 0.018929** | 0.0002668** | 0.0000615** |
| | (0.0009022) | (0.0003301) | (0.0001404) |
| GEcocor | 0.004647** | 0.00486*** | 0.001706*** |
| | (0.0004528) | (0.0002958) | (0.0001003) |
| GEvoac | 0.011723** | 0.0301456*** | 0.027999*** |
| | (0.0001931) | (0.0002054) | (0.0000682) |
| GErequ | 0.014719* | 0.02484*** | 0.0349*** |
| | (0.0007896) | (0.0003418) | (0.0001529) |
| Constant | 7.845032 * | 8.345855* | 8.293294* |
| | (0.7658049) | (1.622502) | (1.539569) |
| Robustness tests | | | |
| Number of observations | 360 | 360 | 360 |
| Wald chi2(7) Prob > chi2 | 0.000 | | 10512 |
| Prob > F | | 0.000 | 0.000 |
| Number of groups | 36 | | |
| Number of instruments | 26 | | |
| F(8,316) | | 1193.27 | |
| Arellano-Bond test for AR(2) | 0.683 | | |
| Hansen test Prob > chi2 | 0.175 | | |

**Source**: Authors computation from STATA15, **Notes**: Economic growth is dependent variable of the study.

*,**, *** denotes 1%, 5% and 10% level of significance respectively, in parenthesis shows the corrected standard errors for system GMM and standard errors for fixed and random effect models.

growth during the period under investigation. It is also revealed that a percentage change in fiscal policy (government expenditure), keeping all other things constant, leads to a 0.20 percent decline in economic growth of SSA. This remained true across the three columns of the finding. This outcome corroborates the works of [62], who found undesirable and significant effect of government final consumption expenditure on economic growth.

The control variables of the study, labor (LBF) and capital (GFCF), as found from the dynamic panel estimation, are positive and significant contributor to the real GDP of SSA countries. We found that a percentage change in the labor results in 0.458 percent increase in economic growth in SSA at 1 percent level of significance, other things remained constant. This result found to be the same across all revealing consistency in the estimation of the results. Furthermore, the two-step system GMM estimation revealed that a one percent change in gross fixed capital formation (which is the proxy of capital in our model), keeping all other things constant, results in a 0.038 percent increase in an economic growth of SSA countries. This finding our study is consistent with the neoclassical economic growth model [32, 63].

Table 8 also presents the interaction of governance indicators and fiscal policy. Accordingly, the interaction of government expenditure with government effectiveness (GEgoef) is found to be negative and significant contributor of economic growth in SSA countries. The interactive coefficient of government expenditure and economic growth is 0.018. It is lower than the absolute value of the direct effect of government expenditure but it points out that there is a need of improving government effectiveness in SSA region. This shows that the improved quality of public and civil services as well as policy formulation and implementation are key inputs of economic growth in SSA countries. Further, the result also presented that the credibility of the government is also very important in the SSA region for economic growth. This finding is similar with the empirical work of [14].

Further, the result of the study revealed that the interaction of fiscal policy with the corruption control of SSA countries (GEcocor) is found to have the coefficient of 0.0046 indicating corruption control (activities against corruption) is the major factor that encourages growth in SSA countries. It is found to be significant at 5 percent level. This finding remained the same across the three column results of our study. The coefficient is very low reflecting that there is a need of improving corruption control in SSA countries. It also revealed that SSA countries need a clean public service instead of corrupted public services. This coincides with the work of [3].

The result of the study further revealed that the interaction of fiscal policy and voice and accountability (GEvoac) also found to positive and significant at 5 percent level of significance as depicted in the three of the result columns. The interactive coefficient is 0.0117 showing that voice and accountability as a governance indicator is very important elements of growth in SSA countries. However, its lower coefficient also tells us that there is a need of improving governance in the SSA region. This reflects that SSA countries need to enhance participation of citizens in decision making and promote the free media where the freedom of association and media is maintained. This finding is corroborating the work of [47].

Finally, the result of our study revealed that interaction of government expenditure and regulatory quality (GErequ) has positive and significant at 5 percent level of significance in three columns of the result. The interactive coefficient is found to be 0.0147 showing that regulatory qualities and policies interacted with government expenditure enhance growth of SSA countries. It further reveals that SSA needs a regulatory quality that promote private sector. Our finding regarding this corroborates with the empirical works of [15, 45].

Finally, our discussion section goes to the bottom section of Table 8 presenting robustness of the model. The two-step system GMM estimated reveals that number of groups are greater than number of instruments (26>36) reflecting that the model is built with collapse option and it is desired as number of instrument is reduced. Further, the estimated model revealed that Arellano-Bond test for AR (2) estimated is 0.683 and it is insignificant reflecting that the estimated model does not suffer from second order serial correlation. This implies that our estimated model has passed a test for zero autocorrelation in first differenced errors. On the other hand, Hansen test is found to be 0.175 showing there is no problem of over identification of the estimated model.

## 5. Conclusion

This study examines the effect of fiscal policy-governance interactions on economic growth of 36 SSA countries over the study periods from 2011 to 2021. In order to do this, dynamic panel data estimation technique through two-step system GMM was employed due to its superiority in generating efficient estimates. The conclusion of the study is that direct economic effect of fiscal policy is negative and significant in SSA countries whereas the interaction of fiscal policy

with governance indicators has positive and significant effect on economic growth. Consequently, before interacting with governance indicators, a percentage change in fiscal policy leads to a 0.20 percent decline in economic growth of SSA countries. Contrary to this, the interactive coefficient of fiscal policy and government effectiveness (0.019) and interactive coefficient of fiscal policy and corruption control (0.0046) are found to be positive and significant. Further, the finding of the study publicized that fiscal policy-voice and accountability interaction coefficient (0.011) and interactive coefficient of fiscal policy-regulatory qualities (0.014) are positively and significantly affecting economic growth of SSA countries.

## 5.1. Managerial implications

The policy implication of the study is that there is a need to improve economic policy credibility and policy formulation as well as its implementation in SSA countries. Further, to bring a better effect of fiscal policy on economic growth in SSA region, strong corruption control that enables the provision of clean public services is very vital. The fact is that trustworthiness can help the government costs to decline. The other implication is that SSA countries need to promote free media and participatory decision making process in a way that promote voice and accountability. Finally, enhancing the positive effect of fiscal policy also requires the improvement of regulatory qualities in SSA countries. Overall suggestion of this study is that SSA countries need to use the fiscal regulatory qualities that promote private sector.

## 5.2. Future recommendations

Since the situation of Africa in general and that of SSA in particular is volatile and dynamic, further studies that examines the effect of fiscal policy across different SSA countries needs to be carried out. The future studies can focus on how institutional quality affects the link between fiscal policy and economic growth of SSA countries. In addition to this, the disparities among low income SSA countries and high income SSA countries need to be investigated. In addition to this, future studies can conduct comparative studies of effect of fiscal policy of SSA countries and other developing countries.

## Supporting information

**S1 Table. Sources and descriptions of the study variables.**
(DOCX)

## Author Contributions

**Conceptualization:** Isubalew Daba Ayana.

**Data curation:** Isubalew Daba Ayana.

**Formal analysis:** Isubalew Daba Ayana.

**Investigation:** Isubalew Daba Ayana.

**Methodology:** Isubalew Daba Ayana.

**Software:** Isubalew Daba Ayana.

**Supervision:** Wondaferahu Mulugeta Demissie, Atnafu Gebremeskel Sore.

**Validation:** Isubalew Daba Ayana.

**Visualization:** Isubalew Daba Ayana.

**Writing – original draft:** Isubalew Daba Ayana.

**Writing – review & editing:** Isubalew Daba Ayana.

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
