## [Decision Letter · Decision Letter 0]

19 Jul 2023

PONE-D-23-18802Fiscal Policy and Economic Growth in Sub Saharan Africa: Do Governance Indicators Matter?PLOS ONE

Dear Dr. Ayana,

Thank you for submitting your manuscript to PLOS ONE. After careful consideration, we feel that it has merit but does not fully meet PLOS ONE’s publication criteria as it currently stands. Therefore, we invite you to submit a revised version of the manuscript that addresses the points raised during the review process.

We look forward to receiving your revised manuscript.

Kind regards,

Syed Ali Raza

Academic Editor

PLOS ONE

Journal Requirements:

Additional Editor Comments:

Considering the reviewers' feedback, I propose that this manuscript should undergo a major revision.

Reviewers' comments:

Reviewer's Responses to Questions

**Comments to the Author**

1. Is the manuscript technically sound, and do the data support the conclusions?

Reviewer #1: No

Reviewer #2: Yes

2. Has the statistical analysis been performed appropriately and rigorously? 

Reviewer #1: Yes

Reviewer #2: Yes

3. Have the authors made all data underlying the findings in their manuscript fully available?

Reviewer #1: Yes

Reviewer #2: No

4. Is the manuscript presented in an intelligible fashion and written in standard English?

Reviewer #1: No

Reviewer #2: Yes

5. Review Comments to the Author

Reviewer #1: The abstract is well-structured and organized in a comprehensive manner.

While the introduction mentions the need for further study on the relationship between fiscal policy and economic growth in sub-Saharan Africa, it doesn't clearly highlight the specific research gap that the study aims to address. The introduction should provide a more focused and concise statement of the problem.

The introduction briefly mentions that the study employs the system GMM estimation method but does not provide a clear rationale for choosing this approach over alternative methods. It must be explained that why the chosen method is appropriate for investigating the research questions and how it addresses the limitations of previous studies.

The review does not explicitly mention the theoretical frameworks or concepts that underpin the studies reviewed. Including a theoretical foundation is important as it would support the framework for understanding the relationships between fiscal policy, governance, and economic growth.

The review focuses primarily on the impact of government expenditure on economic growth, with less emphasis on other aspects of fiscal policy or governance indicators. Provide a more balanced coverage of different dimensions of fiscal policy and governance to enrich the analysis.

Also this section could benefit from the inclusion of additional significant literary articles that could contribute further insights. Furthermore, there is a noticeable scarcity of cited literature pertaining this topic. Consequently, I recommend the incorporation of the following articles, as they align with the aim and objective of the current topic. It is important to cite these articles in your study for enhanced relevance and coverage:

https://doi.org/10.1177/0972150919833484

https://doi.org/10.1108/JFC-01-2021-0003

https://doi.org/10.1177/0972150919846800

https://doi.org/10.1177/09721509221104849

https://doi.org/10.1016/j.gr.2023.05.007

The significant to add Discussion section separately. Further, it is essential to engage in a comprehensive discussion of the findings. Merely interpreting the results is insufficient; authors should delve into the strategies, implications, and rationale underlying the derived outcomes.

To summarize, the conclusion of a study should be divided into three paragraphs: Conclusion, Practical Implications, and Limitations. The first paragraph should summarize the main points of the study, including its objectives, methodology, results, and overall interpretation of findings. The second paragraph should discuss the practical implications of the study and identify key beneficiaries such as policymakers, portfolio managers, the real estate industry, and others. It should also provide strategies for how these beneficiaries can apply the study's findings. The final paragraph should address the limitations of the study and offer suggestions for future researchers. The author should highlight potential areas of exploration that were not covered in the current study but could be explored by future scholars working on a similar topic.

Reviewer #2: The manuscript looks sound methodologically. However, there are few theoretical and conceptual issues.

1. Taking only public expenditure as the proxy of fiscal policy and not taking taxation is unfair. You need to check for the role taxation in economic growth.

2. Relating Governance indicators is a good idea. But institutional framework and institutional quality also plays important role in determine economic growth in many economies. Hence look into this aspect as well.

3. The review of literature is incomplete without a section of review of all the variables in the model you study i.e. relationship between fiscal policy, governance and economic growth together.

4. Substantiate why you have used system GMM. why not PARDL? or CS-ARDL or any other panel cointegration techniques.

5. Use of second generation UNIT ROOT tests are always advisable.

6. Robustness tests are missing from the study.

7. Analysis and policy implications are very weak.

6. PLOS authors have the option to publish the peer review history of their article (what does this mean?). If published, this will include your full peer review and any attached files.

Reviewer #1: No

Reviewer #2: No

---

## [Author Response · Author response to Decision Letter 0]

1 Aug 2023

Date: 30th July, 2023

Response to Reviewers’ Comments

Manuscript No.: PONE-D-23-18802

Title: Fiscal Policy and Economic Growth in Sub Saharan Africa: Do Governance Indicators Matter?

Keywords: Fiscal policy, Governance indicators, two-step system GMM, SSA countries 

Journal: PLOS ONE

Dear Reviewers,

Greetings of the day!

We are thankful to the reviewers for taking the time to assess our manuscript, for their careful reading and for their suggestions and valuable comments which helped us to substantially improve the quality of our paper. In revising the manuscript, we have carefully considered all the raised comments and suggestions. We have attempted to succinctly explain the changes made in reaction to all comments. Our reply to each comment in point-by-point fashion is given in what follows

1. On your concerns regarding abstract

Respected reviewers, thank you for accepting the structure and organization of our abstract. 

2. On your concerns regarding introduction

Dear respected reviewers, we have strongly and positively accepted your comments. Accordingly, we have thoroughly focused on the introduction section during review. We have now addressed the specific research gaps that the study aims to achieve. We have marked it in the marked up copy of our manuscript. Additionally, the introduction section is now focused. It concisely stated the problem of the study. The main reason of selecting system GMM is also discussed in the introduction section. Thank you for your insightful comments. 

3. On your concerns regarding literature review 

Dears, your concerns regarding theoretical literatures are well accepted. We have included theoretical reviews. Moreover, we have raised the evolution of fiscal policy from time of 1930 world great depression to the present institutionalism aspects of governance. We have enriched the both theoretical and empirical literature section. 

Further, we have arranged literatures into two section; theoretical and empirical. Balanced coverage of different dimensions of fiscal policy and governance was provided. We have special regard to your supplementary literatures you supplied to us. We have also cited them in our work to enhance our manuscript. 

4. On your concerns regarding discussion section

Dear reviewers, your concern regarding this section is also right. We appreciate and accepted it positively. Thus, we have added discussion section separately. We have also improved interpretation of the result of our study. In this section, we have provided sufficient discussion of the results.

5. On your concerns regarding conclusion

Following your good comments, the conclusion of a study is divided into three paragraphs. In the first paragraph we have captured conclusion while second paragraph dealt with the practical implications. Third paragraph of this section discussed limitation of the study and directions for future researches. 

6. Some Other parts of your concerns 

To sum up, we would like to appreciate your comments that made our manuscript smarter. Frankly, speaking we learned a lot. This is the way science and knowledge improves. We have included paragraph that discuss robustness check. Further, the major reason why government expenditure is conducted as a proxy of fiscal policy is explained and supported by literature. In developing countries, where the source of growth is mainly sourced from external debt not from tax revenue, the government expenditure can be considered as proxy of fiscal policy. 

 We would like to thank once again all the reviewers for taking the time to review our manuscript, for their relevant remarks and comments, and especially for their specifications which helped us to improve the quality of our paper. 

 Sincerely yours Isubalew Daba

---

## [Decision Letter · Decision Letter 1]

5 Sep 2023

PONE-D-23-18802R1Fiscal Policy and Economic Growth in Sub Saharan Africa: Do Governance Indicators Matter?PLOS ONE

Dear Dr. Ayana,

Thank you for submitting your manuscript to PLOS ONE. After careful consideration, we feel that it has merit but does not fully meet PLOS ONE’s publication criteria as it currently stands. Therefore, we invite you to submit a revised version of the manuscript that addresses the points raised during the review process.

We look forward to receiving your revised manuscript.

Kind regards,

Syed Ali Raza

Academic Editor

PLOS ONE

Journal Requirements:

Additional Editor Comments:

Considering the reviewers' feedback, I propose that this manuscript should undergo a minor revision.

Reviewers' comments:

Reviewer's Responses to Questions

**Comments to the Author**

1. If the authors have adequately addressed your comments raised in a previous round of review and you feel that this manuscript is now acceptable for publication, you may indicate that here to bypass the “Comments to the Author” section, enter your conflict of interest statement in the “Confidential to Editor” section, and submit your "Accept" recommendation.

Reviewer #1: (No Response)

Reviewer #2: All comments have been addressed

2. Is the manuscript technically sound, and do the data support the conclusions?

Reviewer #1: Yes

Reviewer #2: Yes

3. Has the statistical analysis been performed appropriately and rigorously? 

Reviewer #1: Yes

Reviewer #2: Yes

4. Have the authors made all data underlying the findings in their manuscript fully available?

Reviewer #1: No

Reviewer #2: Yes

5. Is the manuscript presented in an intelligible fashion and written in standard English?

Reviewer #1: Yes

Reviewer #2: Yes

6. Review Comments to the Author

Reviewer #1: The author(s) should provide context and comprehensive background information to explain the study’s significance. Also, specify the scope of the study in terms of countries or regions covered and avoid unnecessary repetition of ideas.

In Presentation and discussion of the results (section 6), the results are just statistically interpreted. The author(s) need to illustrate, explain and justify the observed results by discussing the mechanism due which such impact is observed as well as support those results with past studies (the literature covered in Literature review section).

Discussion should be the end section but before Conclusion it should be placed. After applying all test like unit root, robustness checks and other, Discussion is added.

The sequence of the study is very confusing, author(s) are requested to follow the standard headings of the manuscript. Such as:

Chapter 1 Introduction

Chapter 2 Literature review

Literature review can be divided into subheadings

Chapter 3 Methodology and Data

Chapter 4 Data Analysis

Discussion is part of Chapter 4

Chapter 5 Conclusion

The subheadings of this section are 5.1. Managerial Implications; 5.2. Future Recommendations

Reviewer #2: The manuscript described a technically sound scientific research piece with data supporting the conclusions. Experiments have been conducted rigorously, with appropriate controls and sample sizes. The conclusions have been drawn appropriately based on the data presented.

7. PLOS authors have the option to publish the peer review history of their article (what does this mean?). If published, this will include your full peer review and any attached files.

Reviewer #1: No

Reviewer #2: No

---

## [Author Response · Author response to Decision Letter 1]

7 Sep 2023

Date: 7th September, 2023

Response to Reviewers’ Comments

Manuscript No.: PONE-D-23-18802R1

Title: Fiscal Policy and Economic Growth in Sub Saharan Africa: Do Governance Indicators Matter?

Journal: PLOS ONE

Dear Reviewers,

Greetings of the day!

We are thankful to both reviewers for taking the time to assess our manuscript, for their careful reading and for their suggestions and valuable comments which helped us to substantially improve the quality of our paper. In revising the manuscript, we have carefully considered all the raised comments and suggestions. We have attempted to succinctly explain the changes made in reaction to all comments. Our reply to each comment in point-by-point fashion is given in what follows. 

1. On your comments regarding significance and scope of the study

Dear reviewers, thank you for raising this question. We have accepted it positively incorporated the issue. We have specified the scope of the study in terms of countries or regions. Moreover, we have now provided comprehensive background information to explain the study’s significance. We have highlighted this at the end of introduction (page 3 of the manuscript). We have also checked introduction section for unnecessary repetition of ideas. 

2. On your comments in presentation and discussion of the results (section 6 of previous version)

Dear reviewers, thank you once again for the constructive comments you delivered to us. Frankly, our paper benefited a lot from this comment. In the revised manuscript, it is section 4. The discussion of the result is supported by the previous literatures. Discussion of the finding is beyond statistical interpretation of results (section 4.3 of the revised manuscript, page 18-22). 

3. On your comment on regarding sequence of the study

Thank you so much once again for your comments. We have now followed the standard headings of the manuscript. At the end of the introduction, we have added this statement and we have arranged the sequence in line with your comments. 

‘The rest sections of the study are organized as follows. Section 2 presented literature review while methodology and data of the study are conferred in section 3. Further, Section 4 of this study provided data analysis whereas conclusions are discussed in section 5.’ 

Moreover, we have included separate topic for discussion at end of section 4 but before conclusion. 

 We would like to thank once again all the reviewers for taking the time to review our manuscript, for their relevant remarks and comments, and especially for their specifications which helped us to improve the quality of our paper. 

 Sincerely yours, Isubalew Daba 

Corresponding author

---

## [Decision Letter · Decision Letter 2]

8 Oct 2023

Fiscal Policy and Economic Growth in Sub Saharan Africa: Do Governance Indicators Matter?

PONE-D-23-18802R2

Dear Dr. Ayana,

We’re pleased to inform you that your manuscript has been judged scientifically suitable for publication and will be formally accepted for publication once it meets all outstanding technical requirements.

Kind regards,

Syed Ali Raza

Academic Editor

PLOS ONE

Additional Editor Comments (optional):

Considering the reviewers' feedback, I recommend accepting this manuscript.

Reviewers' comments:

Reviewer's Responses to Questions

**Comments to the Author**

1. If the authors have adequately addressed your comments raised in a previous round of review and you feel that this manuscript is now acceptable for publication, you may indicate that here to bypass the “Comments to the Author” section, enter your conflict of interest statement in the “Confidential to Editor” section, and submit your "Accept" recommendation.

Reviewer #1: All comments have been addressed

Reviewer #2: All comments have been addressed

2. Is the manuscript technically sound, and do the data support the conclusions?

Reviewer #1: Yes

Reviewer #2: Yes

3. Has the statistical analysis been performed appropriately and rigorously? 

Reviewer #1: Yes

Reviewer #2: Yes

4. Have the authors made all data underlying the findings in their manuscript fully available?

Reviewer #1: Yes

Reviewer #2: Yes

5. Is the manuscript presented in an intelligible fashion and written in standard English?

Reviewer #1: Yes

Reviewer #2: Yes

6. Review Comments to the Author

Reviewer #1: The authors have implemented the necessary revisions and responded aptly to the inquiries regarding the study, showcasing their dedication and determination to enhance the research's quality.

Reviewer #2: The study presents the results of original research.

Results reported have not been published elsewhere.

Experiments, statistics, and other analyses are performed to a high technical standard and are described in sufficient detail.

Conclusions are presented in an appropriate fashion and are supported by the data.

The article is presented in an intelligible fashion and is written in standard English.

7. PLOS authors have the option to publish the peer review history of their article (what does this mean?). If published, this will include your full peer review and any attached files.

Reviewer #1: No

Reviewer #2: No

---

## [Editor Report · Acceptance letter]

12 Oct 2023

PONE-D-23-18802R2 

Fiscal policy and economic growth in Sub-Saharan Africa: Do governance indicators matter? 

Dear Dr. Ayana:

I'm pleased to inform you that your manuscript has been deemed suitable for publication in PLOS ONE. Congratulations! Your manuscript is now with our production department. 

Kind regards, 

on behalf of

Dr. Syed Ali Raza 

Academic Editor

PLOS ONE